# Microwave Radiometry for the Diagnosis and Monitoring of Inflammatory Arthritis

**DOI:** 10.3390/diagnostics13040609

**Published:** 2023-02-07

**Authors:** Katerina Laskari, Elias Siores, Maria M. Tektonidou, Petros P. Sfikakis

**Affiliations:** 1Rheumatology Unit, 1st Department of Propaedeutic Internal Medicine, Joint Academic Rheumatology Program, University of Athens, Medical School, National & Kapodistrian University of Athens Medical School, 75 Mikras Asias Street, Goudi, 11527 Athens, Greece; 2University of West Attica, 12243 Athens, Greece; 3Institute of Materials Research and Innovation, University of Bolton, Bolton BL3 5AB, UK

**Keywords:** microwave radiometry, synovitis, imaging biomarker, ultrasound, rheumatoid arthritis, osteoarthritis, sacroiliitis

## Abstract

The ability of microwave radiometry (MWR) to detect with high accuracy in-depth temperature changes in human tissues is under investigation in various medical fields. The need for non-invasive, easily accessible imaging biomarkers for the diagnosis and monitoring of inflammatory arthritis provides the background for this application in order to detect the local temperature increase due to the inflammatory process by placing the appropriate MWR sensor on the skin over the joint. Indeed, a number of studies reviewed herein have reported interesting results, suggesting that MWR is useful for the differential diagnosis of arthritis as well as for the assessment of clinical and subclinical inflammation at the individual large or small joint level and the patient level. MWR showed higher agreement with musculoskeletal ultrasound, used as a reference, than with clinical examination in rheumatoid arthritis (RA), while it also appeared useful for the assessment of back pain and sacroiliitis. Further studies with a larger number of patients are warranted to confirm these findings, taking into account the current limitations of the available MWR devices. This may lead to the production of easily accessible and inexpensive MWR devices that will provide a powerful impetus for personalized medicine.

## 1. Emerging need for Sensitive Imaging Techniques for Joint Assessment

Sensitive detection of joint inflammation is of great importance for the diagnosis and successful treatment of joint pathology. Both clinically overt synovitis and clinically silent joint inflammation detected by imaging techniques are considered for a successful treat-to-target strategy. Indeed, clinical examination alone results in a substantial number of false-positive (31%) or false-negative (19%) estimations [1]. It has been demonstrated that approximately one-third of patients suffering from arthralgias have inflammatory changes revealed in ultrasound, and up to half of them will develop inflammatory arthritis in the future [2]. Moreover, up to 44% of RA patients have subclinical inflammatory joint changes, even in the absence of arthralgia and irrespective of whether the definition of remission, using the clinical 28-joint count Disease Activity Score (DAS28) instrument, is met [3]. In addition, clinical instruments, such as DAS28, have considerable test–retest variability [4]. The currently applied treat-to-target therapeutic strategy in RA aims to resolve inflammatory changes early to avoid disease progression and relapses [5] since these changes have been associated with a higher risk of flares, structural damage, and unsuccessful drug tapering [3,6,7]. Thus, medical imaging is becoming increasingly important in assessing outcomes in patients with RA. Obviously, accurate diagnostic methods of inflammatory joint changes adjunct to clinical examination are needed to accomplish this aim. Ultrasonography and magnetic resonance imaging, although sensitive, do not always detect pathology [8]; their interpretation is operator-dependent, and their availability is limited [4]. Thus, the use of alternative, non-invasive, and easily accessible methods for screening in these patients is essential.

Herein, we review the current evidence supporting the use of MWR as another tool for the diagnosis and monitoring of inflammatory arthritis. We aim to summarize the available pilot feasibility studies in order to provide a resource for the use of this technique and its possible future applications. Our systematic search identified 10 studies, shown in Table 1. Definitely, any conclusions on the use of MWR should be interpreted with caution, given the limited currently available data [9].

## 2. Application of Microwaves in Biomedicine

Over the last years, there has been emerging interest in MWR as a tool for detecting tissue pathologies. MWR detects the natural microwave electromagnetic radiation of the body, which is proportional to the absolute temperature. The first microwave radiometers were used on satellites in the 1970s to measure the temperature of the sea surface [19] as well as to remotely monitor climate and weather. Currently, microwave radiation is being used in biomedicine for in-depth tissue temperature measurements [20,21,22,23]. In 1975, Barrett and Myers [24] demonstrated that temperature abnormalities of internal tissue, identified noninvasively by MWR, could be indicative of breast cancer. Except for the diagnosis of cancer [25], several other medical applications have been reported: targeting the brain [26,27,28,29]; detecting atherosclerosis in carotid arteries and, thus, contributing to the primary and secondary prevention of stroke [30,31]; detecting vesicoureteral reflux [32], critical limb ischemia in the diabetic foot [33], and brown adipose tissue activity [34]; the diagnosis of coronavirus disease [35]; microwave-based energy conversion nano-therapy for RA [36] as well as back pain [17,18] and synovial inflammation [1,10,11,12,13,14,15,16,37].

## 3. MWR in Arthritis Assessment

### 3.1. Rationale

The rationale for the use of MWR in inflammatory arthritis is the rise of tissue temperature due to the joint inflammatory process. The local alterations in the synthesis and release of immune mediators in response to injury result in altered biochemical processes and temperature local changes. Indeed, although further studies are warranted, existing evidence suggests that local inflammation in arthritis results in an increased in-depth temperature that can be detected by the MWR sensor, which is placed on the skin over the inflamed joints (Table 1). Since local temperature changes precede structural changes, it is of paramount importance to diagnose inflammatory changes as early as possible, even if signs and symptoms suggestive of inflammation are absent. 

As early as 1949, Horvath and Hollander observed that the temperature of synovial fluid was directly dependent on the relative hyperemia of the synovium [38]. Interest-ingly, the hypervascularization of the synovial membrane, detected as a power doppler (PD) sign by ultrasound, and subclinical persistent synovitis are more strongly associated with disease activity and radiographic changes than clinical and laboratory parameters [3,6]. Likewise, in sacroiliitis, no association has been previously demonstrated between inflammatory sacroiliac changes and inflammatory markers or clinical findings [39].

### 3.2. Technique

RTM-01-RES [Moscow, Russia, and Bolton, UK; http://www.resltd.ru/eng/rtm/ (accessed on 19 January 2023)] (Figure 1) is the commercially available instrument used in most of the studies; however, technical modifications, including different types of sensors and antennas, have been reported, as well as the design of newer devices, such as portable ones [14,22,23,28,29,40,41,42,43]. The RTM-01-RES microwave computer-based system detects the temperature of internal tissues at microwave frequencies, with an accuracy of ±0.2 °C within seconds. The measurement depth depends on the diameter of the sensor used, which touches the surface of the skin. During measurements, the individual lies in a supine position, and the microwave antenna of the device is placed flat on the joint, as shown in Figure 1. The antenna is held in this position for 10 sec. This time is required for the integration of the microwave emission by the receiver and for the conversion of the measured signal to temperature by a microprocessor.

The sensor’s measurement depends on the antenna used since it is a distributed weighted average of temperature within the sensitivity lobe of the antenna [44], as depicted in Figure 1. Therefore, the spatial temperature resolution of the sensor depends strongly on how narrow the sensitivity lobe is. The sensitivity is stronger near the front of the antenna and fades out when moving away from it, having a typical three-dimensional form, as shown in Figure 1.

Although the temperature of object(s) (i.e., the joint) in the vicinity of the antenna is highly weighted and dominates the resulting measurement, the temperature of the surrounding environment, i.e., the air (as well as its electrical properties), impacts the measurement directly (due to being weight-averaged by the antenna) and implicitly (via its impact on the patient’s overall body temperature). To this end, the ambient temperature is of great importance [45]. Notably, others have previously reported that the joint temperature is influenced by seasonal (winter vs. summer) temperature variations [38]. Thus, the examination should be performed at a predefined room temperature. It is recommended that all subjects rest for at least 10 min in an examination room with a steady temperature (21–23 °C) to equilibrate to the ambient temperature before starting the measurements [1]. Moreover, since the measurement result depends on the electromagnetic coupling between the antenna and its surroundings, electronic devices radiating signals within the frequency sensitivity range of the instrument should be silenced, and metallic objects should be moved away.

Regarding the role of intrinsic factors, the geometry and electrical properties of the joint (resulting from biological tissue physicochemical properties) are important. In our previous study, no association was demonstrated between the obtained measured values and patient characteristics such as age, gender, smoking, body mass index, history of diabetes or hypothyroidism, as well as daily prednisolone dose and the use of biological disease-modifying anti-rheumatic drugs [1]. Nevertheless, circadian rhythm, metabolic rate, calorific intake, smoking, physical activity, emotional states such as pain or fear, and atmospheric humidity have all been previously described as parameters that may also have an impact on body temperature [38,46]. For example, small changes in vascular tone can cause large changes in tissue temperature. Thus, there are many uncontrollable factors in our experiment. This may result in significantly different measurements from one patient to another, even from a single joint of one patient at different measurement times. These variations challenge the definition of a standard “normal” temperature [46]. More research is required to determine adjustments for other factors when using MWR to assess arthritis.

Since no reliable data are available on normal absolute temperature cut-offs, the relative temperature, e.g., the difference in temperature (Δt) between two examined areas per joint or between the joint and a control point, is usually calculated. Indeed, Salisbury et al. found that physiological or environmental changes of any individual can shift up or down the heat distribution patterns of a joint, although the actual thermal spread of the heat pattern is constant [46]. This implies a constant relative thermal measurement throughout the day, although the absolute temperature value may vary. Therefore, methods of detecting inflammation using absolute temperature measurement are subject to errors due to these diurnal circadian temperature changes in contrast to those based on relative temperature distribution due to the relative stability of these heat patterns.

In our hands, the temperature difference observed in healthy individuals between the different joint areas was eliminated in inflamed joints, the temperature of which was globally increased [1]. This resulted in lower Δt values in inflamed vs. non-inflamed joints. Overall, the use of MWR showed good intra- and inter-class correlation coefficients, 0.94 and 0.93, respectively, indicating perfect agreement.

### 3.3. Assessment of Large Joints: Knee, Elbow, and Lower Leg

The first joints assessed by MWR were the knees, as early as 1987, when Fraser et al. scanned 52 knees and demonstrated a strong correlation between the microwave thermographic index and the measured clinical and laboratory parameters [10]. Some years later, two studies showed that microwave thermography reflects the clinical change after intra-articular treatment intervention in the rheumatoid knee [11,12]. Based on these findings, in a proof-of-concept study, we tested the hypothesis that increased local temperature detected by MWR reflects subclinical synovial inflammation [13]. Indeed, in 40 RA and 20 osteoarthritis (OA) knees, in the absence of relevant clinical signs, subclinical inflammation detected by ultrasound (fluid effusion and/or PD signal) was characterized by significantly higher absolute (but lower Δt) temperature measurements. Following this study, Ravi et al. used a custom-built radiometer to record the temperature at the lateral and medial compartments of both knees in 41 individuals older than 35 years. In this pilot study, a statistically significant rise in the radiometer temperature was observed at the diseased sites (RA, OA, trauma), unlike the healthy sites [14].

Taking into consideration these observations, in a recent more thorough approach, we tested 82 patients with RA, 26 with OA, as well as healthy controls, and found a stronger inter-rater agreement of MWR with ultrasound-derived PD signs of synovial inflammation (82%) than with clinical examination of the joint (76%) [1]. Interestingly, knees of RA patients with synovitis scored ≥2 or PD activity had significantly lower relative temperature Δt than clinically and sonographically normal knees, while normal knees of patients with RA had comparable temperature values with healthy controls as well as knees with OA. In contrast to RA, the grading of synovial fluid in OA was not associated with the level of the recorded temperature. Among RA patients, MWR had 75% sensitivity and 73% specificity for the detection of knee synovitis scored ≥2, while 80% sensitivity and 82% specificity for PD activity. At the same time, MWR performed well in the discrimination between sonographically abnormal RA knees and knees of patients with OA or healthy controls (83–98% sensitivity, 76–80% specificity). These results are strengthened by a validation assessment of MWR knee measurements in an independent cohort of 31 patients with painful knees, where a knee Δt score of ≤0.2 predicted power Doppler positivity, with 100% positive and negative predictive values.

In the same study, the relative temperature Δt of other large joints (elbow, lower leg) was lower in RA versus controls; nevertheless, MWR performed moderately and similarly to clinical examination in the detection of (teno)synovitis or ultrasound-derived PD sign in these joints.

Subsequently, in a review on imaging diagnostics, Tarakanov et al. referred to the use of MWR in order to determine the depth temperature in the projection of knee joints in 43 children with juvenile idiopathic arthritis as well as 43 healthy children. The authors suggested that clinical visual assessment of temperature data provides an instant idea of thermal asymmetry, which may be used in order to assess diseased versus healthy joints as well as changes in temperature fields observed during the disease course [16].

### 3.4. Assessment of Small Joints: Wrist, Metacarpal Proximal Interphalangeal, and Metatarsophalangeal Joints

The examination of small joints by MWR was systematically assessed by our group [1,15]. Taking into consideration that small joints are mainly affected in RA, in a prospective pilot study, we evaluated 10 patients with active, untreated RA and tested whether the temperature of the small joints, measured by MWR, correlates to disease activity. All patients underwent clinical and laboratory assessments, joint ultrasound, and MWR of the hand joints in serial evaluations up to 3 months after treatment onset [15]. By summing up the relative temperature of 16 joints (second to fifth metacarpal (MCP) and proximal interphalangeal (PIP) joints), bilaterally, we created a thermo-score of small joints, which correlated significantly to the standard disease activity score DAS28, the C-reactive protein serum levels, tender and swollen joint counts, the patient’s visual analog scale, and the standard 7-joint ultrasound score of small joints (US7). Moreover, a statistically significant difference in the thermo-score was observed between patients in high/moderate disease activity/remission versus those in low disease activity/remission or healthy subjects, and individual changes from baseline to the end of follow-up mirrored the corresponding DAS28 changes in the majority of patients.

In a subsequent study, the diagnostic performance of MWR at the single small joint level was tested in 82 patients with RA, 26 patients with OA, and healthy controls [1]. The diagnostic performance of MWR was moderate and similar to clinical examination in the detection of (teno)synovitis or PD in the joints included in the US7 ultrasound score, e.g., the wrist, MCP-2 and -3, PIP-2 and -3, and metatarsophalangeal (MTP)-2 and -5.

### 3.5. Sacroiliac Joint and Low Back Pain Assessment

The first observation with regard to the examination of sacroiliac (SI) joints and lumbar spine with MWR was described in 1987 by Fraser et al. [10]. In ankylosing spondylitis, an almost flat temperature pattern appeared in the lumbar spine in contrast to the normally observed temperature rise at the center of the spine. Nevertheless, the authors were not able to quantify this difference using a numerical form.

Recently, fair sensitivity and specificity of MWR measurements were demonstrated for axial involvement diagnosis in 58 patients with spondyloarthritis [1,37]. In particular, patients with active or chronic sacroiliitis were characterized by significantly lower relative temperature of the sacroiliac (SI) joints (derived by three measurements along the joint area) compared to patients without sacroiliitis or healthy controls (74% sensitivity/78% specificity and 74% sensitivity/72% specificity, respectively). The fact that the presence of bone marrow edema in acute lesions and fat degeneration in chronic lesions were associated with a statistically significant change in the temperature of SI joints may be explained by the presence of inflammatory activity in patients with chronic SI lesions and the better sensitivity of MWR in detecting inflammation. In fact, MWR was able to detect inflammatory SI joint changes in patients with active non-radiological sacroiliitis and in 75% of patients with clinically silent, albeit active, disease in magnetic resonance imaging. MWR measurements showed no correlation to the Bath Ankylosing Spondylitis Disease Activity Index, the patient pain Visual Analogue Scale for SI joints, the New York radiological grading of SI lesions [47], and C-reactive protein levels.

Interestingly, another group has recently investigated the diagnostic role of MWR in patients with low back pain [17,18]. In particular, 48 patients with clinically confirmed acute or sub-acute low back pain and controls were examined by MWR at the level of the spinous processes of the L1 to L5 vertebral bodies along the median, left, and right para-vertebral lines. The area of pathological muscle spasm and/or inflammation in the projection of the vertebral-motor segment was identified by the visualization of thermal asymmetry. The highest internal temperature was observed in patients with the most severe pain and those examined within the first week after the exacerbation. The application of targeted treatment methods resulted in a significant fall in the maximum and normalization of the gradient of internal temperature and, at the same time, a decrease in thermo-asymmetry.

### 3.6. Global Disease Activity Assessment in Patients with Rheumatoid Arthritis

Given the need for new biomarkers, we tried to evaluate global disease activity in RA patients using MWR. Therefore, we assessed small and large joints unilaterally in 56 RA patients who underwent MWR and ultrasound examination simultaneously in the clinically dominant hand/arm and foot/leg in seven small joints (wrist, MCP-2 and -3, PIP-2 and -3, MTP-2 and -5), according to the US7 ultrasound score and three large joints (elbow, knee, ankle) [1]. A statistically significant correlation was shown between the MWR-derived thermo-score of small and large joints and the tender/swollen joint counts, the patient’s and physician’s Visual Analogue Scale, DAS28, inflammatory markers, and the ultrasound scores of synovitis/tenosynovitis. Interestingly, the 10-joint thermo-score was sensitive enough to detect subclinical inflammation since the inter-rater agreement with ultrasound-defined joint inflammation was stronger compared to DAS28 clinical assessment (82% versus 64%). It is of note that 9 out of 11 patients with quiescent disease based on DAS28, albeit active in ultrasound, had a thermo-score indicative of active disease. In addition, a statistically significant difference was observed in the thermo-score among RA disease activity stages. A thermo-score cut-off value of ≤12.1 could discriminate between sonographically-defined RA activity stages with 84% sensitivity/73% specificity. In addition, the thermo-score could discriminate between RA and OA patients with 82% sensitivity/60% specificity and between RA patients and healthy subjects with 74% sensitivity/69% specificity. Finally, we demonstrated the potential role of MWR as a measure of therapeutic response since individual changes in MWR measurements from baseline to follow-up in 20 patients with active disease at baseline mirrored the corresponding DAS28 and ultrasound changes in most patients.

## 4. Current Limitations of MWR for Arthritis Assessment

Considering the MWR operating principle, briefly mentioned in Section 3.2, the main causes of measurement error are (a). the limited spatial resolution of the measurement, (b). the varying geometry and electrical properties of the joint among different patients or of the same patient in different visits, (c). varying environmental temperature, and (d). the distortion of the sensitivity lobe (pattern) of the antenna due to electromagnetic coupling to the environment (metallic objects or electronic devices). Finally, one has to respect the time it takes for the MWR system to make a measurement within the desired temperature accuracy.

Errors due to (a) and (b) have been so far overcome by recording relative (Δt) instead of absolute temperature measurements by pointing (touching) the antenna at two (or more) predefined points of the joint and using an antenna with a sensitivity lobe of appropriate size. Errors due to (c) are compensated by recording relative temperature values as well as by maintaining a fixed temperature in the examination room and by having all subjects rest in the room to equilibrate to the ambient temperature before starting the measurements. Errors due to (d) are minimized by switching off or moving away cellphones, computers, smartwatches, and other electronic devices emitting radio frequency signals, as well as metallic objects.

Another issue to consider is the lack of studies demonstrating the high sensitivity and specificity of MWR in the examination of joints other than the knee; no reports are available on the assessment of shoulder or hip joints.

Finally, the reference method used in the reported studies to indicate inflammation is musculoskeletal ultrasound and/or clinical examination and/or magnetic resonance imaging for low back pain; none of them is the gold standard for joint assessment. In this respect, no studies have investigated the association between MWR findings and the quantification of inflammation in tissue biopsies.

## 5. Comparison of Joint MWR to Other Imaging Modalities

Compared to other imaging methods of the joint, such as ultrasound, X-rays, computed tomography, and magnetic resonance imaging, MWR does not display morphological alterations. However, the measuring procedure and the result interpretation are simple and objective, and no special education is required. Overall, it is a fast, cheap, and safe technique [16,48]. In particular, unlike X-ray and computed tomography, it is a passive screening method and does not emit radiation. Unlike magnetic resonance imaging, MWR represents a low-cost, portable imaging technology that allows examination over short time intervals. Although ultrasound depicts the inflamed synovium, including synovial vascularity, while magnetic resonance imaging allows the assessment of osteitis beyond the bony cortex, the use of these techniques is operator-dependent and requires special knowledge, which is not always feasible in routine outpatient visits. In contrast, MWR could be of value even in situations where the rheumatologist’s physical examination cannot be performed. Both joint ultrasound and magnetic resonance imaging are used for the diagnosis of early inflammatory changes such as subclinical synovial inflammation and bone erosions, which are not accompanied by clinical signs or symptoms and may not be visible in X-rays. Likewise, MWR may be used as a method to detect early inflammatory changes since the already reported evidence points towards its utility for the diagnosis of subclinical inflammation.

Finally, compared to the commonly applied tool of infrared thermography for the diagnosis of inflamed joints, which detects electromagnetic radiation from superficial tissues, the use of microwaves determines the temperature distribution from superficial to deeper body structures, where measurements are influenced to a lesser extent by extrinsic factors [49]. Various studies using thermography reassure the observation that temperature measurements may be used as a diagnostic tool in RA as well as a tool to assess disease activity; it shows a stronger correlation to joint ultrasound than clinical findings [50,51]. Tarakanov et al. [16] observed a pattern match between the temperature field of the skin and that of deeper tissues under normal conditions; nevertheless, some decades earlier, Horvath and Hollander showed that clinical activity in RA correlated more strongly with the intra-articular temperature compared to the superficial skin temperature [38]. Likewise, the amount of heat “felt” on the skin surface over a joint was not always related to joint temperature.

To this respect, more research and comparative studies are required to evaluate the advantages and disadvantages and the exact role of MWR as a diagnostic, prognostic, and outcome measurement tool in the clinical management of inflammatory arthritis compared to other imaging modalities. At the same time, the combined use of MWR with other imaging methods that complement each other may offer a more comprehensive overall assessment of joint inflammation and, hence, disease activity in patients with inflammatory arthritis when compared to the use of one imaging modality alone.

## 6. Conclusions and Future Perspectives

There is growing evidence supporting the role of MWR as an imaging biomarker that detects clinical as well as subclinical inflammation at the joint and patient levels. In addition, MWR seems to discriminate between RA disease activity stages, to distinguish between patients and controls, and may serve as a measure of therapeutic response. At the same time, MWR appears useful for the assessment of back pain and sacroiliitis. As an additional diagnostic tool, MWR shows higher agreement with musculoskeletal ultrasound findings than clinical assessment. In this respect, MWR should not substitute clinical examination. ΜWR may be used as an adjunctive method to improve the sensitivity/specificity of joint assessment. Certainly, validation studies of a large number of patients are warranted.

Moreover, refinement of this technique in the future, which depends significantly on new equipment and computers, may increase its accuracy as well as its sensitivity and specificity. Advancements in MWR can be made by improving several aspects of it [21,22,23,28,29,40,41,43,44]. The antenna is of critical importance, and a more targeted design, resulting in higher spatial resolution and larger operating frequency bandwidth to allow for faster measurements, would be very useful. Frequency and bandwidth selection in the antenna and amplifying chain would also help to isolate external interference and achieve better calibration. Fast measurement capability, along with an inertial (or another type of) antenna’s position and attitude tracking, would enable the generation of thermal maps, providing significantly more information than single measurements. Finally, machine learning tools such as convolutional neural network models could be used to process the temperature measurements and create reference thermal maps with enhanced resolution, taking into account the anatomical geometry and properties of the part of the body under examination. Deep neural network models could be applied to support advanced diagnostic methodologies and clinical studies and to monitor the effects of treatment. To the best of our knowledge, medical radiometry is at the research stage, with no formal approval for particular diagnostic purposes. The production of easily accessible and inexpensive devices will provide a powerful impetus for personalized medicine in the future.

## Figures and Tables

**Figure 1 diagnostics-13-00609-f001:**
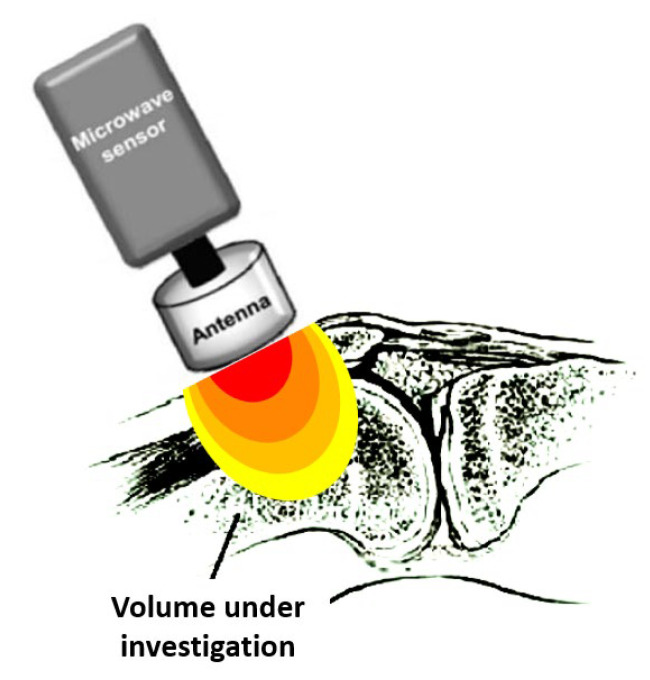
The measured temperature is the average temperature within the volume in the vicinity of the antenna, weighted according to its sensitivity pattern (lobe). The temperature of the dark (red) area volume is of more weight in the measurement than that of the volume in the light (yellow) area. Modified from Zampeli et al. [13].

**Table 1 diagnostics-13-00609-t001:** Published evidence on the use of microwave radiometry for the assessment of inflammatory arthritis and low back pain.

Reference	N of Patients	Assessed Joints	Main Findings
Fraser et al. 1987 [10]	− 44 knees of RA patients, 8 healthy knees − Patients with AS and healthy controls (N unknown)	Knee, lumbar spine	− The microwave thermographic index correlates strongly to clinical and laboratory parameters. − In AS, an almost flat temperature pattern appears in contrast to the normal temperature rise to the center of the lumbar spine.
MacDonald et al. 1994 [11]	24 RA, 2 PsA, 2 overlap of RA and SLE	Knee	− A weak correlation of thermal measurements to parameters of disease activity is observed. − Microwave thermography reflects the clinical change after intra-articular treatment intervention.
Blyth et al. 1998 [12]	82 RA	Knee	Microwave thermography reflects the clinical change after intra-articular treatment intervention.
Zampeli et al. 2013 [13]	20 RA, 10 OA	Knee bilaterally	MWR detects subclinical synovial inflammation observed in joint ultrasound.
Ravi et al. 2019 [14]	41 RA or OA or trauma or healthy cases	Knee	A statistically significant rise in the radiometer temperature is observed at the diseased, unlike the healthy, sites.
Pentazos et al. 2018 [15]	10 RA	Small joints: 2nd to 5th MCP and PIP joints, bilaterally	The MWR-derived thermo-score of 16 small joints correlates significantly to clinical and laboratory parameters. The thermos-score discriminates between RA disease activity stages as well as between patients and controls and reflects disease activity changes after treatment.
Laskari et al. 2020 [1]	82 RA, 26 OA, 58 SpA, 48 healthy controls	− Small and large joints in the clinically dominant hand/arm and foot/leg: wrist, MCP-2,3, PIP-2,3, MTP-2,5, elbow, knee, ankle − Sacroiliac joints	− Knees of RA patients with synovitis scored ≥2 or PD activity have a significantly lower relative temperature Δt than clinically and sonographically normal knees, while normal knees of patients with RA have comparable temperature values with healthy controls, as well as knees with OA. − MWR performance in the other large as well as small joints is moderate and comparable to clinical examination. − Patients with active or chronic sacroiliitis have a lower relative temperature of sacroiliac joints compared to patients without sacroiliitis or healthy controls. − In RA, the MWR-derived thermo-score of 7 small and 3 large joints correlates significantly to clinical and laboratory parameters, discriminates between RA disease activity stages as well as between patients and controls, and reflects disease activity changes after treatment.
Tarakanov et al. 2021 [16]	43 JIA and 43 healthy children	Knee	The visual assessment of temperature data reveals a thermal asymmetry, which may be used to assess diseased versus healthy joints as well as changes in temperature fields observed during the disease course.
Tarakanov et al. 2021 [17]	48 with low back pain	Lumbar spine	The internal temperature is highest in most severe pain and in patients examined within the first week after the exacerbation.
Tarakanov et al. 2022 [18]	55 with low back pain	Lumbar spine	− The abnormal area is identified by thermal asymmetry visualization, which is defined by temperature fields. − The application of targeted treatment results in a significant fall in the maximum and normalization of the gradient of internal temperature and, at the same time, a decrease in thermo-asymmetry.

RA: rheumatoid arthritis; AS: ankylosing spondylitis; OA: osteoarthritis; PsA: psoriasis arthritis; SLE: systemic lupus erythematosus; SpA: spondyloarthritis; MCP: metacarpophalangeal; PIP: proximal inter-phalangeal; MTP: metatarsophalangeal; MWR: microwave radiometry; JIA: juvenile idiopathic arthritis.

## Data Availability

Not applicable.

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
