# Peer review of "Microwave Radiometry for the Diagnosis and Monitoring of Inflammatory Arthritis"

_diagnostics, 2023, doi:10.3390/diagnostics13040609_

Round 1

Reviewer 1 Report (Previous Reviewer 2)

authors have adequately responded to first set of reviews.

interesting and worthy paper.

Reviewer 2 Report (Previous Reviewer 1)

The authors have addressed all of the reviewers' comments and suggestions. I believe that the manuscript may be published in its present form.

This manuscript is a resubmission of an earlier submission. The following is a list of the peer review reports and author responses from that submission.

Round 1

Reviewer 1 Report

The ms is well written and structured containing interesting information, reflecting its title correctly. I have two main suggestions that I think should be taken into account before article publication.

I believe the introduction section does not correctly mention the work of the research groups working in the field that have until very recently provided advances in both methods and techniques in MR  biomedical applications. I would propose to substitute some older references with more recent ones. Please see for example:

P. Momenroodaki, W. Haines, M. Fromandi, and Z. Popovic, “Noninvasive internal body temperature tracking with near-field microwave radiometry,” IEEE Transactions on Microwave Theory and Techniques, vol. 66, no. 5, pp. 2535–2545, 2017.

E. Fedoseeva, G. Shchukin, and I. Rostokin, “Calibration of a Tri-Band Microwave Radiometric System with Background Noise Compensation,” Measurement Techniques, vol. 63, no. 4, pp. 301–307, 2020.

S. Vesnin, A. K. Turnbull, J. M. Dixon, and I. Goryanin, “Modern microwave thermometry for breast cancer,” J. Mol. Imag. Dynamic, vol. 7, no. 136, pp. 10–1109, 2017.

B. Osmonov et al., “Passive Microwave Radiometry for the Diagnosis of Coronavirus Disease 2019 Lung Complications in Kyrgyzstan,” Diagnostics, vol. 11, no. 2, 2021, doi: 10.3390/diagnostics11020259.

E. Groumpas, M. Koutsoupidou, I. S. Karanasiou, C. Papageorgiou and N. Uzunoglu, "Real-Time Passive Brain Monitoring System Using Near-Field Microwave Radiometry," in IEEE Transactions on Biomedical Engineering, vol. 67, no. 1, pp. 158-165, Jan. 2020, doi: 10.1109/TBME.2019.2909994.

E. I. Groumpas, M. Koutsoupidou and I. S. Karanasiou, "Biomedical Passive Microwave Imaging and Sensing: Current and future trends [Bioelectromagnetics]," in IEEE Antennas and Propagation Magazine, vol. 64, no. 6, pp. 84-111, Dec. 2022, doi: 10.1109/MAP.2022.3210860.

K. Tisdale, A. Bringer and A. Kiourti, "Development of a Coherent Model for Radiometric Core Body Temperature Sensing," in IEEE Journal of Electromagnetics, RF and Microwaves in Medicine and Biology, vol. 6, no. 3, pp. 355-363, Sept. 2022, doi: 10.1109/JERM.2021.3137962.

K. Tisdale, A. Bringer and A. Kiourti, "A Core Body Temperature Retrieval Method for Microwave Radiometry When Tissue Permittivity is Unknown," in IEEE Journal of Electromagnetics, RF and Microwaves in Medicine and Biology, vol. 6, no. 4, pp. 470-476, Dec. 2022, doi: 10.1109/JERM.2022.3171092.

I also think a small description of optimal measurement protocols and acquisition procedures stating possible limitations would be useful to those working in the field.

Reviewer 2 Report

An interesting, well written review. One major and several minor comments

1. Major. Authors present an optimistic view about the use of microwave thermography for diagnosing and monitoring arthritis. HOWEVER all of the biomedical papers they discuss are very small, apparently unblinded - seem to be clinical feasibility studies. In fact, for clinical use in diagnosing arthritis, to be acceptable to clinicians (let alone be approved for reimbursement) the specificity and sensitivity of the method must be known under typical use conditions. That would require much larger and better conducted (blinded) studies than those described here.  It is not possible to predict the outcome of such studies. 

Moskowitz (Am. J. Radiol. 140:591-594 1983) wrote a strong warning against overselling new imaging methods, referring to the disastrous experience with thermography for breast cancer detection. I see signs in this paper of the same overselling that led to the debacles with thermographic and breast imaging, including microwave thermography.  Would the authors *please* add appropriate cautions, point out that the work they review is clinical feasibility stage.

I recommend (but I do not require) citing the Moskowitz paper that outlines a progressive series of studies from proof of concept state (what we seem to have here) to studies that would plausibly demonstrate clinical efficacy.

Less major

1. Paper loosely uses the ambiguous term "significantly"- should state "statistically significant" instead and quote the effect size in a standards form such as Cohen's d.

2. Authors state that the effective depth of sensing of microwave thermography is 2-6 cm. Not true, exactly. Due to the antenna reciprocity theorem, the received signal is a weighted average of energy emitted at various depths, weighted by an exponential factor that emphasizes temperatures near the surface.(Cheever and Foster, IEEE Trans Biomed Eng 39(6) 563-8, 1992).

3. "Overall, the use of MWR showed good intra- and inter-class correlation coefficients, 0.94 and 0.93, respectively, indicating perfect agreement". Hardly "perfect agreement". Please show the effect size (Cohen's d), which is more informative than correlation coefficients. 

4. Would the authors please discuss: what would be the medical requirements for sensitivity of diagnosis of arthritis, given the costs and benefits of true/false positives? For typical prevalence rates in the target population, what would be the false positive and false negative rates?

Would it be an adjunctive method of diagnosis or primary diagnostic tool?

5. I find the suggestion of the method for at home diagnosis/screening implausible and unwarranted at this stage of development, until we have some understanding of the usefulness of the method in a clinical setting. 

But please, don't oversell the method, which would could lead to the same disaster as happened with thermography for detection of breast cancer (which is still being practiced by chiropractors and alternative medicine practitioners, even as mainstream radiologists have long since lost interest in the method). That could have all sorts of bad consequences for what at present is an interesting and worthy proposal.

My comments can be accommodated by minor revision, adding a few paragraphs and modifying a few sentences. Otherwise an interesting and worthy review. 

Reviewer 3 Report

The authors review the recent studies on the microwave radiometry for the diagnosis and monitoring of inflammatory arthritis. However, the  paper is not well organized and presented. The authors must improve the paper by considering the following points.

1. clearly state the research motivation.

2. section 1 is too short, does section 1 is introduction section? If yes, a comprehensive introduction and more relevant references should be provided.

3. Figure 1 is difficult to read, the image resolution must be improved.

4. Sections 3, 4 and 5 can be reorganized 

5. More relevant research references especially recent publications should be provided in the reference section. 

6. Research findings should be clearly stated.